# Cross-City Latent Space Alignment for Consistency Region Embedding

Meng Chen [1]  Hongwei Jia [1]  Zechen Li [1]  Wenzhen Jia [2]  Kai Zhao [3]  Hongjun Dai [1]  Weiming Huang [4]

## Abstract

Learning urban region embeddings has substantially advanced urban analysis, but their typical focus on individual cities leads to disparate embedding spaces, hindering cross-city knowledge transfer and the reuse of downstream task predictors. To tackle this issue, we present Consistency Region Embedding (CoRE), a unified framework integrating region embedding learning with cross-city latent space alignment. CoRE first embeds regions from two cities into separate latent spaces, followed by the alignment of latent space manifolds and fine-grained individual regions from both cities. This ensures compatible and comparable embeddings within aligned latent spaces, enabling predictions of various socioeconomic indicators without ground truth labels by migrating knowledge from label-rich cities. Extensive experiments show CoRE outperforms competitive baselines, confirming its effectiveness for cross-city knowledge transfer via aligned latent spaces.

## 1. Introduction

Learning vector representations (embeddings) for small urban units (regions) using various urban sensory data sources, including points of interest and human trajectories, has become a common practice in urban computing and analytics (Chen et al., 2024; Huang et al., 2023; Li et al., 2023; Jia et al., 2024; Mai et al., 2023; Zhang et al., 2024; Chen et al., 2025b). These representations have been proven to be valuable for various urban downstream tasks, including regional land use analysis, crime rate prediction, and house price forecasting, demonstrating significant potential for real-world applications in urban planning and management

[1]School of Software, Shandong University, Jinan, China [2]Business School, Shandong Normal University, Jinan, China [3]Walmart AI Lab, California, USA [4]Department of Physical Geography and Ecosystem Science, Lund University, Sweden. Correspondence to: Hongjun Dai <dahogn@sdu.edu.cn>, Weiming Huang <W.Huang@leeds.ac.uk>.

*Proceedings of the 42nd International Conference on Machine Learning*, Vancouver, Canada. PMLR 267, 2025. Copyright 2025 by the author(s).

(Chen et al., 2025a; Li et al., 2022; Ning et al., 2024; Zhang et al., 2023).

From a holistic perspective, as illustrated in Figure 1(a), learning region representations typically involves two stages: 1) regions from city $X$ are projected into a latent space to generate representations ($\mathbf{Z}^X$), and 2) these representations are fed into downstream task predictors ($dec_X$) to produce analytical results. However, the second stage requires sufficient ground truth data to train $dec_X$, which is often difficult to acquire in practice. Many studies focus on cities like New York City, where ground truth labels are readily available with open data projects, while other cities remain understudied (Zhang et al., 2022).

To address this issue, we aim to answer the following question in the paper: **can we align the latent spaces between different cities so that we can leverage the knowledge from one city to facilitate the analysis in other cities?** Technically, this hinges on whether the latent embedding spaces learned from urban sensory data in different cities are comparable. The straightforward answer is that the latent spaces learned separately for different cities are typically incomparable and unaligned, making it difficult to transfer knowledge through them. This incomparability issue is shown in Figure 2(a), where latent spaces learned by MVURE (Zhang et al., 2021) are projected onto the same 2D plane using t-Distributed Stochastic Neighbor Embedding (t-SNE) (Maaten & Hinton, 2008). The triangles and dots represent regions in two cities (Chengdu and Xi'an), with yellow, green, and red indicating low, medium, and high GDP levels, respectively. The two spaces are clearly different, indicating a meaningless direct comparison of two cities.

In this paper, we aim to achieve the knowledge transfer between cities by reusing task predictors ($dec_X$). These predictors, trained in a label-rich city, can be reused in another city without ground truth labels, enabling cross-city predictions. Recent studies have realized this through unsupervised translation (Yabe et al., 2019; 2020; Haikal et al., 2022), where they transform region representations learned in one city to the other, similar to translating words across languages, as shown in Figure 1(b). Specifically, they follow a two-stage paradigm: intra-city region embedding learning and cross-city translation. First, region embedding meth-

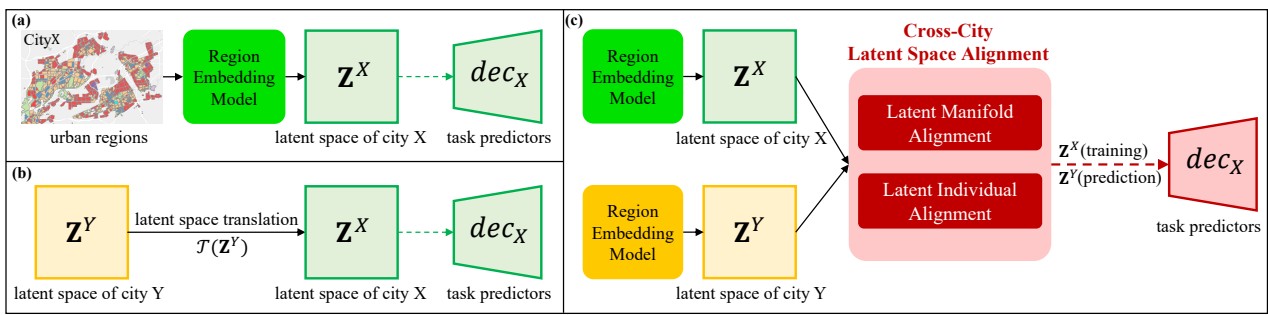

*Figure 1.* Illustration of: (a) learning and using region representations within a single city, (b) existing latent space translation methods that requires a direct mapping $\mathcal{T}$ from $\mathbf{Z}^Y$ to $\mathbf{Z}^X$, and (c) our CoRE method that integrates region embedding learning with cross-city alignment to produce compatible representations across cities, ensuring that semantically similar regions lie close in the aligned spaces.

ods generate region representations ($\mathbf{Z}^X$ and $\mathbf{Z}^Y$) within individual cities ($X$ and $Y$), where the two latent spaces are incompatible. Then they manually construct anchoring pairs of region representations across cities using fixed correspondence rules (e.g., urban hierarchical structures) to learn transformation functions ($\mathbf{Z}^X \approx \mathcal{T}(\mathbf{Z}^Y)$). This process translates the region representations of one city ($Y$) to the space of another ($X$), enabling the reuse of task predictors ($dec_X$) in city $Y$ based on $\mathcal{T}(\mathbf{Z}^Y)$.

While these methods have shown promise, they suffer from two major shortcomings:

1) **Disconnected Embedding and Translation**: In two-stage approaches, the decoupled learning paradigm creates a fundamental limitation: the frozen pre-trained embeddings in the translation stage force cross-city mapping to operate on rigid latent spaces. This rigidity constrains the translation function to suboptimal linear projections between selected anchor pairs, failing to adapt to the inherent non-linear manifold structures of urban feature spaces. Therefore, developing a one-stage method that jointly learns and aligns region embeddings through a synergistic optimization process is needed.

2) **Reliance on Hand-Crafted Correspondence**: Previous methods depend on fixed, hand-crafted correspondence (e.g., matching regions across cities by check-in counts), which can be incomplete or biased - especially since user check-ins reflect social media biases (Wang et al., 2016; Li et al., 2024). An adaptive cross-city alignment method free of manual rules is needed to address this limitation.

In this paper, we propose a one-stage **Co**nsistency **R**egion **E**mbedding method (CoRE), which integrates region embedding learning with cross-city latent space alignment to produce compatible and comparable region representations across cities, as shown in Figure 1(c). Recognizing the challenge of achieving cross-city alignment without human-crafted rules and labels, CoRE accomplishes this through two mutually reinforcing pathways: latent manifold align-

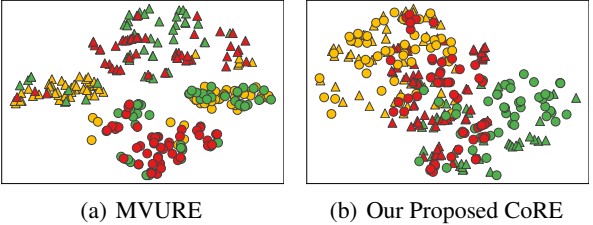

(a) MVURE    (b) Our Proposed CoRE

*Figure 2.* Visualization of the pre-trained region embeddings. We project embeddings into 2D via t-SNE, where triangles and dots denote regions in Chengdu and Xi'an, respectively. Colors (yellow, green, red) indicate low, medium, and high GDP levels. Panel (a) shows the MVURE (Zhang et al., 2021) embeddings, with clearly separate city spaces, while panel (b) shows our CoRE embeddings, where semantically similar regions are closely aligned.

ment, and individual region alignment. Specifically, CoRE consists of three components. First, we employ a graph attention network as the encoder to generate city-specific region representations in distinct latent spaces using human mobility data. Second, we construct virtual parallel common anchors in both spaces as bridges and align the data manifolds using the idea of relative representations. Third, we develop a cross-city attention pipeline to transfer pairwise region correlations between two spaces, ensuring embedding consistency at the individual region level.

As illustrated in Figure 2(b), CoRE aligns the latent spaces of two cities such that semantically similar regions across cities are positioned closely. This makes CoRE useful for urban embedding learning, which enables knowledge transfer between different cities trained on similar urban data types. The key contributions of this paper are as follows:

- We propose CoRE, a one-stage consistency region embedding method that integrates region embedding learning with cross-city latent space alignment in a unified framework. This approach ensures compatibility and comparability of region representations from different cities within

an aligned latent space.

- We achieve cross-city alignment of latent spaces through two interdependent pathways: latent manifold alignment and latent individual alignment, addressing both global and individual perspectives. Our method accounts for the complex interplay between cities without relying on explicit anchoring pairs of region representations across cities based on human-crafted rules.

- We conduct comprehensive experiments across three downstream tasks using datasets from three cities. The results clearly show the advantages of CoRE over baseline methods. Data and source codes are available at: https://github.com/AIMUrban/CoRE.

## 2. PROBLEM FORMULATION

**Definition 2.1** (**Urban Region**). A city is divided into $n$ disjoint urban regions $\mathcal{R} = \{r_1, r_2, \ldots, r_n\}$, where each region $r_i$ is defined by a set of boundary points specified by their latitude and longitude coordinates.

**Definition 2.2** (**Region Representations**). For a region $r_i^X$ in city $X$, its representation is encoded as a $d$-dimensional vector $\mathbf{Z}_i^X$. The representations of all regions in city $X$ are collectively represented as a $(N_X \times d)$ matrix $\mathbf{Z}^X$, where $N_X$ denotes the number of regions in the city.

**Definition 2.3** (**Consistency Region Embedding across Cities**). Our objective is to generate compatible region representations for cities $X$ and $Y$ by aligning their latent spaces. This alignment ensures that semantically similar regions from both cities are positioned closely within the shared latent space. Achieving consistent region embedding for cities $X$ and $Y$ makes their representations compatible and comparable. Thus, downstream task predictors trained with region representations in city $X$, where ample ground truth data is available, can be directly applied to make accurate predictions in city $Y$.

## 3. Method

In this section, we first introduce the base region embedding model, which independently learns region representations for each city. We then propose a cross-city latent manifold alignment component designed to align the data manifolds of two latent spaces without requiring external supervision. Finally, we present a cross-city latent individual alignment component that enhances fine-grained alignment of individual region representations through a self-supervised task.

### 3.1. Base Region Embedding Model

Human mobility within a city reflects the relationships between its different regions, making it a popular data source for learning region representations (Yao et al., 2018; Liu

et al., 2024). To this end, we first construct a base region embedding model for an individual city. This model captures human mobility patterns between regions and derives latent region representations using a self-supervised graph embedding approach.

#### 3.1.1. MOBILITY GRAPH CONSTRUCTION

Given a human mobility dataset $\mathcal{M} = \{m_1, m_2, \cdots\}$, where each trip $m_i = (r_o, r_d)$ represents a movement from an origin region $r_o$ to a destination region $r_d$, we construct a mobility graph $\mathcal{G} = \{\mathcal{V}, \mathcal{E}\}$ for each city. Here, $\mathcal{V} = \{r_i\}_{i=1}^N$ represents the $N$ regions as nodes, and the edge weights $\mathcal{E} = \{w_{ij}\}_{i,j=1}^N$ are derived from $\mathcal{M}$. Specifically, the transition frequency (edge weight) from region $r_i$ to region $r_j$ is calculated as:

$$w_{ij} = \text{count}(r_i \to r_j) / \sum_{k=1}^N \text{count}(r_i \to r_k), \quad (1)$$

where $\text{count}(r_i \to r_j)$ denotes the number of trips from $r_i$ to $r_j$. This normalization ensures that each edge weight reflects the proportion of trips from a given origin to each possible destination. Using this approach, we construct two separate mobility graphs, $\mathcal{G}^X$ and $\mathcal{G}^Y$, for cities $X$ and $Y$, respectively.

#### 3.1.2. GRAPH EMBEDDING

Based on the mobility graph $\mathcal{G}$, we employ a graph attention network (GAT) (Veličković et al., 2017) to learn the latent representation of each node (region). The input vectors for the $N$ regions are initialized as $\{\mathbf{E}_i \in \mathbb{R}^{1 \times d}\}_{i=1}^N$. The GAT layer computes the feature representation $\mathbf{Z}_i \in \mathbb{R}^{1 \times d}$ for region $r_i$ as follows:

$$\mathbf{Z}_i = \frac{1}{H} \sum_{h=1}^H \mathbf{Z}_i'^{(h)}, \quad \mathbf{Z}_i'^{(h)} = \sum_{r_j \in \mathcal{N}_i} \alpha_{ij}^{(h)} \mathbf{E}_j \mathbf{W}^{(h)},$$

$$\alpha_{ij}^{(h)} = \exp(s_{ij}^{(h)}) / \sum_{r_k \in \mathcal{N}_i} \exp(s_{ik}^{(h)}), \quad (2)$$

$$s_{ij}^{(h)} = \text{LeakyReLU}([\mathbf{E}_i \mathbf{W}^{(h)} \| \mathbf{E}_j \mathbf{W}^{(h)} \| w_{ij} \mathbf{W}^{e(h)}] \cdot \mathbf{a}^{(h)\mathrm{T}}),$$

where $\mathbf{W}^{(h)} \in \mathbb{R}^{d \times d}$, $\mathbf{W}^{e(h)} \in \mathbb{R}^{1 \times d}$, and $\mathbf{a}^{(h)} \in \mathbb{R}^{1 \times 3d}$ are learnable parameters of the $h$-th attention head, $H$ is the number of attention heads, $\|$ denotes concatenation, and $\mathcal{N}_i$ represents the neighbors of region $r_i$.

#### 3.1.3. INTRA-CITY REGION EMBEDDING LOSS

To effectively learn region representations, we utilize a self-supervised task that reconstructs mobility patterns among regions within an individual city. Specifically, we predict the destination region $r_j$ given $r_i$ using the region representations $\mathbf{Z}$. For a specific region $r_i$, the distribution of the

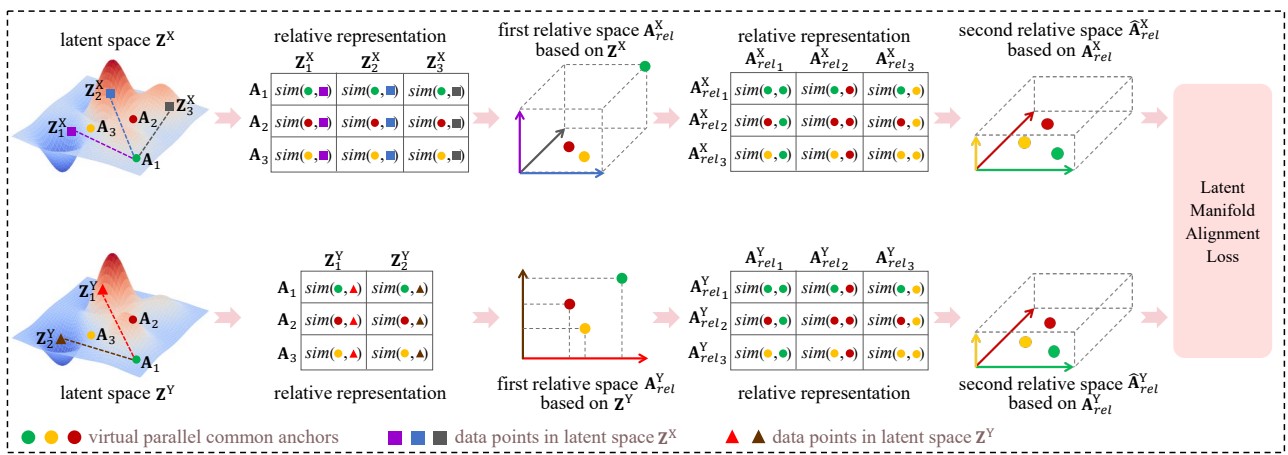

*Figure 3.* The framework of cross-city latent manifold alignment. Three common anchors $\mathbf{A}_1$, $\mathbf{A}_2$, and $\mathbf{A}_3$ (colored dots) are randomly generated in both latent spaces. For a given anchor $\mathbf{A}_i$, we compute its similarity w.r.t. the region representations on the data manifold of $\mathbf{Z}^X$ and $\mathbf{Z}^Y$, yielding vectors ($\mathbf{A}_{rel_i}^X$ and $\mathbf{A}_{rel_i}^Y$) of dimensionality 3 and 2, respectively. Each dimension is treated as coefficients in a coordinate system defined by all region representations, which are orthogonal in this example only for visualization purposes. Similarly, we compute the second relative representations $\hat{\mathbf{A}}_{rel}^X/\hat{\mathbf{A}}_{rel}^Y$ w.r.t. $\mathbf{A}_{rel}^X/\mathbf{A}_{rel}^Y$, which are now in the same coordinate system. Finally, we minimize the difference between $\hat{\mathbf{A}}_{rel}^X$ and $\hat{\mathbf{A}}_{rel}^Y$ to ensure the alignment of the latent manifolds of $\mathbf{Z}^X$ and $\mathbf{Z}^Y$.

destination region $r_j$ is computed as:

$$\hat{p}\left(r_j \mid r_i\right) = \exp(\mathbf{Z}_i \cdot \mathbf{Z}_j{}^\mathrm{T}) / \sum_{k=1}^{N} \exp(\mathbf{Z}_i \cdot \mathbf{Z}_k{}^\mathrm{T}). \quad (3)$$

Using the human mobility data $\mathcal{M}$, we define the intra-city region embedding (RE) loss as:

$$\mathcal{L}^{\mathrm{RE}} = -\frac{1}{|\mathcal{M}|} \sum_{\left(r_i, r_j\right) \in \mathcal{M}} \log \hat{p}\left(r_j \mid r_i\right). \quad (4)$$

This approach reconstructs actual mobility patterns based on the graph structure, enabling us to derive the region representations $\mathbf{Z}^X$ and $\mathbf{Z}^Y$ for both cities.

### 3.2. Cross-City Latent Manifold Alignment

The region embedding model in Section 3.1 generates city-specific region representations in distinct latent spaces. Aligning these separate latent spaces for cross-city comparison poses significant challenges, particularly due to the absence of labeled paired data or correspondence annotations, as well as the differing numbers of regions across cities. To address these challenges, we propose a cross-city latent manifold alignment method. This method leverages the relative representations of virtual common anchors as a shared reference system for both cities. By replacing the original absolute representations with a relative reference system, defined by the similarity between region representations and common anchor points, we ensure that the region representations of both cities are learned within a unified reference framework (see Figure 3).

#### 3.2.1. RELATIVE REPRESENTATIONS OF PARALLEL COMMON ANCHORS

Due to the varying number of data points (regions) in each latent space, directly measuring the similarity between the data manifolds of two latent spaces is challenging. To address this, we propose using $N_a$ randomly generated virtual parallel anchor points, $\mathbf{A} \in \mathbb{R}^{N_a \times d}$, which remain consistent across the latent spaces of different cities. These anchors serve as bridges for comparison, where $N_a$ is the number of anchors, and their dimensionality matches that of the region representations.

To represent the manifold of the latent space using these anchors, we adopt the concept of relative representations (Moschella et al., 2023). Specifically, each anchor is represented relative to all region representations in the latent space. We define a generic similarity function $sim : \mathbb{R}^d \times \mathbb{R}^d \to \mathbb{R}$ to capture the relationship between the anchors and the region representations, yielding a scalar score $\mathbf{A}_{rel_{ij}} = sim(\mathbf{A}_i, \mathbf{Z}_j)$ for two absolute representations $\mathbf{A}_i$ and $\mathbf{Z}_j$. Given the region representations $\mathbf{Z}_1^X, \cdots, \mathbf{Z}_{N_X}^X$ in the latent space $\mathbf{Z}^X$, the relative representation of $\mathbf{A}_i$ is

$$\mathbf{A}_{rel_i}^X = (sim(\mathbf{A}_i, \mathbf{Z}_1^X), \cdots, sim(\mathbf{A}_i, \mathbf{Z}_{N_X}^X)), \quad (5)$$

where cosine similarity is chosen as the similarity function due to its invariance to angle-preserving transformations. Similarly, the relative representation of $\mathbf{A}_i$ with respect to the latent space $\mathbf{Z}^Y$ is computed as:

$$\mathbf{A}_{rel_i}^Y = (sim(\mathbf{A}_i, \mathbf{Z}_1^Y), \cdots, sim(\mathbf{A}_i, \mathbf{Z}_{N_Y}^Y)). \quad (6)$$

Based on Equations (5) and (6), we compute the relative

representations of all anchors in matrix form, projecting each anchor point $\mathbf{A}_i \in \mathbb{R}^d$ into the region sets $\mathcal{R}^X \subset \mathbb{R}^{N_X}$ and $\mathcal{R}^Y \subset \mathbb{R}^{N_Y}$ in the latent spaces $\mathbf{Z}^X$ and $\mathbf{Z}^Y$. This is formally represented as:

$$\mathbf{A}_{rel}^X = \mathbf{A} \cdot \mathbf{Z}^{X\,\mathrm{T}}, \quad \mathbf{A}_{rel}^Y = \mathbf{A} \cdot \mathbf{Z}^{Y\,\mathrm{T}}, \tag{7}$$

where $\mathbf{A}_{rel}^X \in \mathbb{R}^{N_a \times N_X}$ and $\mathbf{A}_{rel}^Y \in \mathbb{R}^{N_a \times N_Y}$, in which each entry quantifies the similarity between an anchor and a region within the city. The samples in $\mathbf{A}_{rel}^X$ and $\mathbf{A}_{rel}^Y$ are rescaled to unit norm, capturing the intrinsic shape of the data by considering only the relative angles between points.

We use the parallel common anchors as bridges and project $\mathbf{A}_{rel}^X$ and $\mathbf{A}_{rel}^Y$ into a relative space based on the common anchor set $\mathcal{A}$. Each sample is represented as a function of the common anchor set, computed as:

$$\hat{\mathbf{A}}_{rel}^X = \mathbf{A}_{rel}^X \cdot \mathbf{A}_{rel}^{X\,\mathrm{T}}, \quad \hat{\mathbf{A}}_{rel}^Y = \mathbf{A}_{rel}^Y \cdot \mathbf{A}_{rel}^{Y\,\mathrm{T}}, \tag{8}$$

where $\hat{\mathbf{A}}_{rel}^X \in \mathbb{R}^{N_a \times N_a}$ and $\hat{\mathbf{A}}_{rel}^Y \in \mathbb{R}^{N_a \times N_a}$ are also rescaled to unit norm. At this stage, through two rounds of relative representations, $\hat{\mathbf{A}}_{rel}^X$ and $\hat{\mathbf{A}}_{rel}^Y$ are aligned within the same reference system.

### 3.2.2. LATENT MANIFOLD ALIGNMENT LOSS

For well-aligned latent spaces $\mathbf{Z}^X$ and $\mathbf{Z}^Y$, the difference between $\hat{\mathbf{A}}_{rel}^X$ and $\hat{\mathbf{A}}_{rel}^Y$ remains small, as the latent manifolds of the two spaces are similar. Conversely, significant discrepancies between the latent manifolds result in large differences in $\hat{\mathbf{A}}_{rel}^X$ and $\hat{\mathbf{A}}_{rel}^Y$. To ensure the alignment of the latent manifolds, we minimize the difference between $\hat{\mathbf{A}}_{rel}^X$ and $\hat{\mathbf{A}}_{rel}^Y$ and define the latent manifold alignment (LMA) loss as:

$$\mathcal{L}^{\mathrm{LMA}} = \frac{1}{N_a^2} \|\hat{\mathbf{A}}_{rel}^X - \hat{\mathbf{A}}_{rel}^Y\|_2^2, \tag{9}$$

where $\|\cdot\|_2$ denotes the $L_2$ norm. This difference is solely determined by the parallel common anchors in the latent spaces and is independent of the number of regions in each city. Moreover, the latent manifold alignment does not rely on human-defined rules for creating anchoring pairs of region representations across cities.

### 3.3. Cross-City Latent Individual Alignment

In addition to aligning the latent manifolds of two spaces, we aim to achieve more fine-grained alignment of individual region representations. Effective alignment should bring semantically similar regions from different cities closer together, which is crucial for cross-city knowledge transfer. To this end, we propose a self-supervised task that enhances alignment from an individual perspective. The core idea is to utilize cross-space attention mechanisms to transfer pairwise region correlations between the two latent spaces:

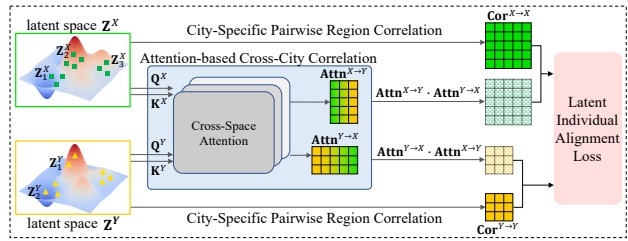

*Figure 4.* Cross-city latent individual alignment.

from $\mathbf{Z}^X$ to $\mathbf{Z}^Y$ and vice versa. Specifically, if we can cycle through $\mathbf{Z}^X \to \mathbf{Z}^Y \to \mathbf{Z}^X$ (or $\mathbf{Z}^Y \to \mathbf{Z}^X \to \mathbf{Z}^Y$) and ensure that the pairwise region correlations in the original and cycled spaces closely match, it indicates that the cross-city pairwise relations among regions are well preserved. This serves as strong evidence that the two latent spaces are further aligned at the individual level.

This process is analogous to machine translation between two languages (e.g., English to Spanish). For instance, translating words from English to Spanish and then back to English should preserve the semantic relationships between the original English words. If these relationships remain consistent before and after translation, we can be confident that the translation process is reliable at the individual-word level. Similarly, our cross-space attention mechanism ensures that the pairwise relationships between regions are preserved across cities, enhancing the alignment of individual region representations.

The proposed latent individual alignment method consists of several key steps, as illustrated in Figure 4. First, we compute city-specific pairwise region correlations to create ground truth labels. Next, we employ a cross-attention mechanism to generate cross-city correlation matrices from $\mathbf{Z}^X$ to $\mathbf{Z}^Y$ and from $\mathbf{Z}^Y$ to $\mathbf{Z}^X$. Finally, we use these cross-city pairwise correlation matrices to reconstruct intra-city pairwise region correlations and minimize the reconstruction differences based on the ground truth signals.

### 3.3.1. CITY-SPECIFIC PAIRWISE CORRELATIONS

Using the region representations in the latent spaces $\mathbf{Z}^X$ and $\mathbf{Z}^Y$, we compute the intra-city pairwise region correlation matrices $\mathbf{Cor}^{X \to X} \in \mathbb{R}^{N_X \times N_X}$ and $\mathbf{Cor}^{Y \to Y} \in \mathbb{R}^{N_Y \times N_Y}$ as ground truth labels:

$$\mathbf{Cor}^{X \to X} = \mathbf{Z}^X \cdot (\mathbf{Z}^X)^{\mathrm{T}}, \quad \mathbf{Cor}^{Y \to Y} = \mathbf{Z}^Y \cdot (\mathbf{Z}^Y)^{\mathrm{T}}. \tag{10}$$

### 3.3.2. ATTENTION-BASED CROSS-CITY PAIRWISE CORRELATION

To measure cross-city region correlations, we employ a cross-attention mechanism that facilitates interaction be-

tween region representations from the two latent spaces. The cross-attention module consists of two parallel attention operations: queries $\mathbf{Q}^X$ and keys $\mathbf{K}^X$ are derived from latent space $\mathbf{Z}^X$, while queries $\mathbf{Q}^Y$ and keys $\mathbf{K}^Y$ are derived from latent space $\mathbf{Z}^Y$. By computing the dot product of queries and keys, we obtain the attention-based cross-city pairwise region correlation matrices $\mathbf{Attn}^{X\rightarrow Y} \in \mathbb{R}^{N_X \times N_Y}$ and $\mathbf{Attn}^{Y\rightarrow X} \in \mathbb{R}^{N_Y \times N_X}$. This process is defined as:

$$
\begin{aligned}
\mathbf{Attn}^{X\rightarrow Y} &= \mathbf{Q}^X \cdot (\mathbf{K}^Y)^{\mathrm{T}}/\sqrt{d}, \\
\mathbf{Attn}^{Y\rightarrow X} &= \mathbf{Q}^Y \cdot (\mathbf{K}^X)^{\mathrm{T}}/\sqrt{d}, \\
\mathbf{Q}^X &= \mathbf{Z}^X\mathbf{W}^q, \quad \mathbf{K}^X = \mathbf{Z}^X\mathbf{W}^k, \\
\mathbf{Q}^Y &= \mathbf{Z}^Y\mathbf{W}^q, \quad \mathbf{K}^Y = \mathbf{Z}^Y\mathbf{W}^k,
\end{aligned}
\tag{11}
$$

where $\mathbf{W}^q \in \mathbb{R}^{d\times d}$ and $\mathbf{W}^k \in \mathbb{R}^{d\times d}$ are trainable parameters.

### 3.3.3. LATENT INDIVIDUAL ALIGNMENT LOSS

Using the two cross-city pairwise correlation matrices, we reconstruct the city-specific pairwise region correlations through a cyclical approach. We then minimize the reconstruction differences based on the ground truth city-specific pairwise region correlations and define the latent individual alignment (LIA) loss as:

$$
\begin{aligned}
\mathcal{L}^{\mathrm{LIA}} =\, &\frac{1}{N_X^2}\|\mathbf{Cor}^{X\rightarrow X} - \mathbf{Attn}^{X\rightarrow Y} \cdot \mathbf{Attn}^{Y\rightarrow X}\|_2^2 \\
+\, &\frac{1}{N_Y^2}\|\mathbf{Cor}^{Y\rightarrow Y} - \mathbf{Attn}^{Y\rightarrow X} \cdot \mathbf{Attn}^{X\rightarrow Y}\|_2^2.
\end{aligned}
\tag{12}
$$

By minimizing this loss, the pairwise relative positions of regions are preserved. This holds true for both intra-city and inter-city region pairs, strongly indicating that the individual region representations from the two cities are positioned reasonably within the aligned latent spaces.

### 3.4. Model Training

Our proposed CoRE is a one-stage approach that simultaneously learns region representations for two cities and aligns their latent spaces. We combine the objectives into the overall loss function:

$$
\mathcal{L} = (\mathcal{L}_X^{\mathrm{RE}} + \mathcal{L}_Y^{\mathrm{RE}}) + \alpha\mathcal{L}^{\mathrm{LMA}} + \beta\mathcal{L}^{\mathrm{LIA}},
\tag{13}
$$

where $\mathcal{L}_X^{\mathrm{RE}}$ and $\mathcal{L}_Y^{\mathrm{RE}}$ denote the intra-city region embedding losses for cities $X$ and $Y$, respectively. The terms $\alpha$ and $\beta$ are hyperparameters that adjust the weights of the LMA loss and the LIA loss, respectively. This objective function can be optimized using stochastic gradient descent. Once optimized, the two latent spaces $\mathbf{Z}^X$ and $\mathbf{Z}^Y$ are aligned within the same coordinate system, positioning semantically similar regions from different cities closer together. This alignment facilitates effective cross-city knowledge transfer and enables the reuse of task predictors across cities.

## 4. Experiments

In this section, we evaluate the proposed CoRE on three cross-city urban prediction tasks with real-world datasets and compare its performance against various baselines.

### 4.1. Data

We use real-world datasets from three cities: Xi'an (XA), Chengdu (CD), and Beijing (BJ). Each dataset comprises urban sensory data (including region data and human mobility data) and downstream label data (i.e., urban socioeconomic indicator data). The details of each data type are introduced in Appendix A.1.

### 4.2. Experiment Settings

#### 4.2.1. MODEL PARAMETERS

In the base region embedding model, we utilize a two-layer GAT as the encoder with an embedding dimension $d$ set to 128 and the attention heads $H$ set to 8. For cross-city latent manifold alignment, we initially generate 1000 anchors and further assess the impact of varying anchor counts in our experiments. The optimization of the model is conducted using the Adam optimizer, with a learning rate configured at $1 \times 10^{-3}$.

#### 4.2.2. BASELINES

To evaluate CoRE, we compare it against unsupervised translation methods and domain adaptation techniques. Among the translation methods, we primarily benchmark against the approach proposed by Yabe et al. (2020). This method first generates city-specific region representations and then employs three strategies (rank-based, hierarchical stochastic, and hierarchical batch) to construct anchoring embedding matrices across cities. These matrices are aligned using Orthogonal Procrustes and Affine alignment techniques, resulting in six baselines: RP (Rank-based + Procrustes), RA (Rank-based + Affine), HSP (Hierarchical Stochastic + Procrustes), HSA (Hierarchical Stochastic + Affine), HBP (Hierarchical Batch + Procrustes), and HBA (Hierarchical Batch + Affine). Furthermore, we include MMD (Saito et al., 2018), Adv (Ganin et al., 2016), and CrossTReS (Jin et al., 2022) as domain adaptation baselines, along with CARPG (Yang et al., 2023) as a knowledge transfer baseline. We also introduce a lower-bound baseline, Individual, which generates city-specific region representations without any cross-city alignment. Detailed descriptions of these baselines are provided in Appendix A.2.

#### 4.2.3. EVALUATION METRICS

We conduct cross-city region-level prediction of socioeconomic indicators using region representations in aligned

*Table 1.* Performance comparison on three downstream tasks with the XA and CD datasets, where the best results are represented in bold and the performance improvements of CoRE are compared with the second best results marked by underlined.

| Method | XA($X$)/CD($Y$) | | | | | | CD($X$)/XA($Y$) | | | | | |
|---|---|---|---|---|---|---|---|---|---|---|---|---|
| | GDP | | Population | | Carbon | | GDP | | Population | | Carbon | |
| | MAE | MAPE | MAE | MAPE | MAE | MAPE | MAE | MAPE | MAE | MAPE | MAE | MAPE |
| Individual | 475.84 | 20.39 | 1694.82 | 7.92 | 480.05 | 21.69 | 247.9 | 10.61 | 978.15 | 11.92 | 176.19 | 4.56 |
| RP | 189.75 | 11.91 | 664.31 | 3.62 | 145.11 | 10.51 | 191.44 | 6.19 | 672.58 | 5.39 | 148.11 | 2.53 |
| RA | 197.47 | 12.58 | 654.83 | 3.84 | 143.99 | 10.49 | 187.07 | 6.08 | 614.32 | 5.27 | 143.04 | 2.09 |
| HSP | 189.48 | 11.66 | 674.31 | 3.74 | 143.46 | 10.13 | 192.22 | 6.43 | 697.59 | 5.71 | 144.49 | 2.1 |
| HSA | 183.68 | 11.51 | 642.17 | 3.71 | 140.73 | 9.98 | 182.52 | 5.6 | 622.43 | 5.33 | 141.06 | 1.97 |
| HBP | 196.14 | 11.52 | 682.71 | 3.82 | 149.6 | 10.33 | 228.25 | 8.65 | 808.61 | 7.84 | 152.76 | 2.96 |
| HBA | 196.67 | 11.53 | 661.88 | 3.61 | 153.71 | 9.77 | 187.7 | 5.68 | 653.88 | 5.53 | 142.91 | 2.07 |
| MMD | 205.48 | 12.97 | 767.26 | 4.8 | 153.38 | 9.9 | 208.17 | 7.07 | 776.82 | 6.61 | 148.05 | 2.51 |
| Adv | 204.92 | 12.79 | 742.62 | 4.13 | 153.8 | 10.04 | 216.67 | 7.81 | 779.52 | 5.75 | 153.77 | 3.01 |
| CrossTReS | 201.06 | 12.71 | 698.49 | 3.98 | 164.81 | 11.21 | 195.13 | 6.81 | 689.83 | 5.69 | 142.96 | 2.08 |
| CARPG | 188.08 | 11.89 | 650.62 | 3.85 | 133.82 | 9.66 | 189.81 | 6.14 | 655.39 | 5.58 | 144.99 | 2.21 |
| CoRE | **178.93** | **11.4** | **620.15** | **3.42** | **127.45** | **9.42** | **170.25** | **5.06** | **580.98** | **4.88** | **136.56** | **1.77** |
| Improvement (%) | 2.59 | 0.96 | 3.43 | 5.26 | 4.76 | 2.48 | 6.72 | 9.64 | 5.43 | 7.4 | 3.19 | 10.15 |

latent spaces. The training data consists of region representations ($\mathbf{Z}^X$) and socioeconomic indicator labels ($\{\mathbf{y}_i^X\}_{i=1}^{N_X}$), including GDP, population, and carbon emissions, from the source city ($X$). The test data comprises region representations ($\mathbf{Z}^Y$) and corresponding labels ($\{\mathbf{y}_i^Y\}_{i=1}^{N_Y}$) from the target city ($Y$). First, a downstream task predictor $dec_X$ is trained using the source city's representations $\mathbf{Z}^X$ and socioeconomic labels $\{\mathbf{y}_i^X\}_{i=1}^{N_X}$. The predictive performance is evaluated by applying the trained $dec_X$ to the target city's representations $\mathbf{Z}^Y$. Ridge regression is employed as the task predictor. We use mean absolute error (MAE) and mean absolute percentage error (MAPE). Lower values of MAE and MAPE indicate higher predictive accuracy.

### 4.3. Performance Comparison with Baselines

Table 1 presents the cross-city socioeconomic indicator prediction performance of various methods on the XA and CD datasets, where $X$ denotes the source city and $Y$ denotes the target city. Each method is evaluated over five runs, and the average values of MAE and MAPE are reported. Additional results for other city pairs are provided in Appendix A.3. From these results, we have the following observations:

- The Individual method performs the worst, as it fails to align latent spaces across cities. This underscores the necessity of cross-city alignment to effectively reuse task predictors trained in data-rich cities. In contrast, the latent space translation methods (RP, RA, HSP, HSA, HBP, and HBA) achieve better performance. These methods leverage urban hierarchical structures to construct effective correspondences between regions, enabling the transformation of target city region representations into the source city's latent space.

- The MMD and CrossTReS methods exhibit poor performance, as they primarily focus on aligning feature distributions and fail to adequately capture inter-region relationships. Additionally, disparities in the number of regions between cities further compromise their alignment effectiveness. The CARPG exhibits limited performance due to its inability to explicitly align feature distributions between cities. The Adv method also underperforms compared to other methods, which we attribute to the limitations of unsupervised learning. In the absence of labeled data from the target city, the alignment process cannot be directly validated or optimized based on predictive performance, potentially leading to incomplete or inaccurate alignment.

- Our proposed CoRE method achieves the best performance with two key contributions: 1) it aligns both the latent manifolds of embedding spaces and individual region representations across cities, and 2) the cross-city alignment enhances the quality of region embedding learning. The superiority of CoRE is further validated by paired t-tests (Hull, 1993), which confirm that its improvements over the baselines are statistically significant ($p$-value < 0.01).

To further validate the effectiveness of our approach, we visualize the region representations learned by HBA and CoRE for two cities by projecting them onto 2D planes using t-SNE (Maaten & Hinton, 2008). Detailed results are provided in Appendix A.4. The visualizations demonstrate that CoRE effectively aligns individual region representations across cities, achieving fine-grained alignment of latent spaces from both global and local perspectives.

We also investigate whether the cross-city alignment compromises within-city prediction performance, and if so, to what extent. The detailed analysis in Appendix A.5 confirms that this compromise is minor. Therefore, CoRE maintains strong within-city predictive accuracy while successfully

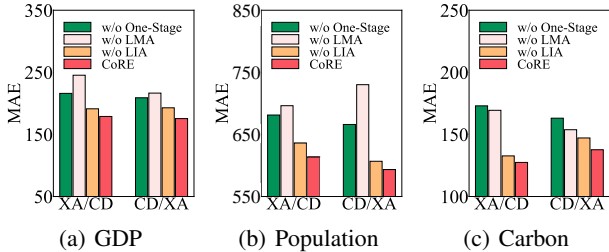

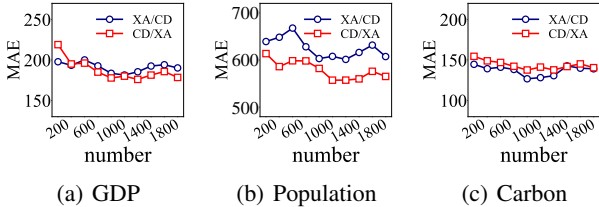

*Figure 5.* Performance comparison of different variants.

*Figure 6.* Effect of the number of virtual parallel anchors.

enabling cross-city alignment.

### 4.4. Ablation Study and Parameter Analysis

#### 4.4.1. STUDY OF DIFFERENT VARIANTS

To evaluate the contribution of each component in CoRE, we conduct an ablation study by designing three variants: 1) **CoRE w/o One-Stage**: The one-stage paradigm is separated into two stages. First, region representations $\mathbf{Z}^X$ and $\mathbf{Z}^Y$ are learned using the intra-city region embedding loss. Then, $\mathbf{Z}^X$ and $\mathbf{Z}^Y$ are projected into two vector spaces via affine layers and aligned using $\mathcal{L}^{\text{LMA}}$ and $\mathcal{L}^{\text{LIA}}$, without updating the parameters of the base region embedding model. 2) **CoRE w/o LMA**: The cross-city latent manifold alignment component is removed, and only the losses $\mathcal{L}^{\text{RE}}$ and $\mathcal{L}^{\text{LIA}}$ are used. 3) **CoRE w/o LIA**: The cross-city latent individual alignment component is removed, and only the losses $\mathcal{L}^{\text{RE}}$ and $\mathcal{L}^{\text{LMA}}$ are used.

The results of CoRE and its variants in terms of MAE for XA/CD and CD/XA pairs are shown in Figure 5. Additional results for other city pairs are provided in Appendix A.6. We observe that both CoRE w/o LMA and CoRE w/o LIA underperform compared to CoRE. The former struggles to align embedding spaces from a global perspective, while the latter lacks the ability to finely align individual region representations across cities. Additionally, CoRE w/o One-Stage exhibits inferior performance due to its inability to adaptively adjust region representations based on the effectiveness of cross-city latent space alignment.

#### 4.4.2. STUDY OF PARAMETER SENSITIVITY

The parameter $N_a$ represents the number of virtual parallel anchors used in latent manifold alignment. To evaluate its impact on model performance, we vary $N_a$ from 200 to 2000 in increments of 200. Figure 6 presents the MAE results for the XA/CD and CD/XA pairs, with similar trends observed for the other four city pairs. Our analysis reveals that CoRE performs poorly when $N_a = 200$, indicating that an insufficient number of anchors hinders effective manifold alignment. Increasing $N_a$ from 200 to 1000 leads to a significant improvement in performance, suggesting

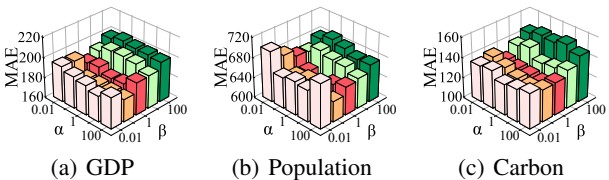

*Figure 7.* Effect of weights $\alpha$ and $\beta$.

that a larger number of anchors enhances the model's ability to align latent manifolds. However, further increasing $N_a$ beyond 1000 results in performance degradation, likely due to overfitting or increased computational complexity.

The parameters $\alpha$ and $\beta$ control the influence of different components in the total loss function. To investigate their impact, we vary their values within the range {0.01, 0.1, 1, 10, 100}. Figure 7 illustrates the MAE results for the XA/CD pair, demonstrating that CoRE achieves optimal performance when both $\alpha$ and $\beta$ are set to 1. Deviating from this balance, either by increasing or decreasing $\alpha$ and $\beta$, leads to suboptimal performance.

## 5. Conclusion

In this paper, we introduce CoRE, a framework that integrates region embedding learning with cross-city latent space alignment to produce compatible and comparable consistency region representations across cities. CoRE comprises three key modules: 1) an intra-city region embedding module that generates city-specific region representations from human mobility data, 2) a latent manifold alignment module that aligns data manifolds of two spaces using relative representations, 3) a latent individual alignment module that ensures precise alignment of individual regions across cities through a cross-attention mechanism. Experimental results on data from three cities demonstrate that CoRE outperforms state-of-the-art methods in cross-city prediction tasks. Importantly, our approach eliminates the need for manual inter-city correspondence rules, making it broadly applicable to unsupervised latent space alignment tasks.

## Impact Statement

Learning effective representations for urban regions (neighborhoods) has become an increasingly popular approach for various urban analyses, such as inferring land use and population distribution. These representations are typically learned in a self-supervised manner, without requiring task-specific ground truth data. However, making task-specific predictions still relies on ground truth data to train predictors, which is often scarce or unavailable in practice. In this work, we propose a cross-city latent space alignment method that enables the transfer of learned patterns across cities. This allows task predictors trained in a city with ample ground truth data to be reused in another city with no such data, making it possible to use region representations in a fully label-free manner.

## Acknowledgments

This work was supported in part by the National Natural Science Foundation of China under Grant No. 61906107 and 42101421, the Key Scientific and Technological Innovation Project of Shandong Province under Grant No. 2024CXGC010113 and 2024CXG010213, and the Young Scholars Program of Shandong University. W.H. was supported by the Knut and Alice Wallenberg Foundation.

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

# A. Appendix

## A.1. Data

We use real-world datasets from three cities: Xi'an (XA), Chengdu (CD), and Beijing (BJ). Each dataset comprises three types of data, namely region data, human mobility data and socioeconomic indicator data. The details of each data type are as follows:

1) **Region data** is obtained from Beijing City Lab[1]. Each city is segmented into various regions, which serve as basic spatial units defined by the road network. The number of regions in XA, CD, and BJ datasets are 1306, 1056, and 1311, respectively.

2) **Human mobility data** includes one-month taxi trips from three cities, with each trip providing the latitude and longitude of both the origin and destination (Jiang et al., 2023). The XA, CD, and BJ datasets contain 559729, 384618, and 78945 trips, respectively.

3) **Socioeconomic indicator data** includes GDP data (Zhao et al., 2017), population data (Bondarenko et al., 2020), and carbon emission data (Oda et al., 2018) of the three cities, which are used in downstream tasks. Since the data types are raster and the regions are irregular, we first map the pixels to the regions and then use the sum values as the final socioeconomic indicators for those regions.

## A.2. Description of Baselines

We compare our proposed CoRE with unsupervised latent space translation methods and domain adaptation techniques. Among the translation methods, we primarily benchmark against the approach proposed by (Yabe et al., 2020). This method first generates city-specific region representations and then employs three strategies (rank-based, hierarchical stochastic, and hierarchical batch) to construct anchoring embedding matrices across cities. These matrices are aligned using Orthogonal Procrustes and Affine alignment techniques, resulting in the following baselines:

- **RP** (Rank-based Anchoring + Procrustes): Establishes one-to-one correspondences by sorting regional ranks within each city, forming anchoring pairs that are aligned using the Orthogonal Procrustes method.

- **RA** (Rank-based Anchoring + Affine): Similar to RP but employs Affine alignment instead of Orthogonal Procrustes.

- **HSP** (Hierarchical Stochastic Anchoring + Procrustes): Randomly selects anchoring pairs within each hierarchical level across cities and aligns them using the Orthogonal Procrustes method.

- **HSA** (Hierarchical Stochastic Anchoring + Affine): Uses Affine alignment instead of Orthogonal Procrustes, as in HSP.

- **HBP** (Hierarchical Batch Anchoring + Procrustes): Computes mean vectors of region representations at each hierarchical level, constructs anchoring matrices, and aligns them using the Orthogonal Procrustes method.

- **HBA** (Hierarchical Batch Anchoring + Affine): Replaces Orthogonal Procrustes with Affine alignment, differing from HBP.

Additionally, we include three domain adaptation methods and one knowledge transfer method for comparison:

- **MMD** (Saito et al., 2018): Utilizes Maximum Mean Discrepancy (MMD) to minimize representation distribution disparities between source and target cities for effective alignment.

- **Adv** (Ganin et al., 2016): Implements domain adversarial training to make region representations indistinguishable between cities, enabling distribution alignment through model parameter adjustments.

- **CrossTReS** (Jin et al., 2022): Employs MMD to reduce the distribution discrepancies in representations between source and target cities. Furthermore, it constructs edge embeddings by concatenating the features of paired regions. An edge classification task is implemented to enhance the alignment of feature distribution.

---

[1]https://www.beijingcitylab.com/data-released-1/

*Table 2.* Performance comparison on three downstream tasks, where the best results are represented in bold and the performance improvements of CoRE are compared with the second best results marked by underlined.

| Method | BJ($X$)/CD($Y$) | | | | | | CD($X$)/BJ($Y$) | | | | | |
|---|---|---|---|---|---|---|---|---|---|---|---|---|
| | GDP | | Population | | Carbon | | GDP | | Population | | Carbon | |
| | MAE | MAPE | MAE | MAPE | MAE | MAPE | MAE | MAPE | MAE | MAPE | MAE | MAPE |
| Individual | 245.46 | 16.06 | 1170.74 | 9.89 | 361.29 | 25.57 | 321.48 | 12.63 | 997.96 | 6.62 | 240.71 | 6.56 |
| RP | 210.89 | 11.89 | 1065.99 | 8.81 | 253.03 | 15.53 | 187.07 | 8.15 | 803.38 | 6.24 | 198.69 | 3.79 |
| RA | 173.46 | 8.5 | 720.68 | 5.58 | 189.94 | 13.84 | 161.96 | 6.98 | 731.01 | 5.48 | 181.74 | 3.09 |
| HSP | 198.11 | 11.45 | 1074.02 | 8.56 | 255.89 | 15.98 | 163.98 | 6.92 | 743.87 | 5.55 | 197.54 | 3.83 |
| HSA | 176.49 | 8.57 | 685.37 | 4.19 | 170.52 | 12.46 | 158.64 | 6.55 | 726.02 | 5.35 | 189.95 | 3.68 |
| HBP | 202.74 | 11.04 | 1092.02 | 8.82 | 250.42 | 14.86 | 187.39 | 8.13 | 843.17 | 6.53 | 187.69 | 3.63 |
| HBA | 152.18 | 6.63 | 707.67 | 4.2 | 182.68 | 13.62 | 184.49 | 7.18 | 759.61 | 5.68 | 186.49 | 3.49 |
| MMD | 179.28 | 8.58 | 904.79 | 6.75 | 230.04 | 14.4 | 199.39 | 8.66 | 873.23 | 6.64 | 197.82 | 3.82 |
| Adv | 227.23 | 13.92 | 940.57 | 7.27 | 330.95 | 23.53 | 193.39 | 8.39 | 825.77 | 6.33 | 186.31 | 3.45 |
| CrossTReS | 188.66 | 9.61 | 902.92 | 6.45 | 222.77 | 14.19 | 179.59 | 7.84 | 755.34 | 5.71 | 183.55 | 3.32 |
| CARPG | 173.24 | 8.07 | 813.84 | 6.56 | 195.91 | 14.18 | 191.25 | 8.54 | 741.44 | 5.53 | 187.13 | 3.58 |
| CoRE | **148.55** | **6.22** | **665.16** | **4.07** | **148.49** | **10.09** | **141.96** | **5.79** | **676.87** | **4.73** | **179.25** | **2.95** |
| Improvement (%) | 2.39 | 6.18 | 2.95 | 2.86 | 12.92 | 19.02 | 10.51 | 11.6 | 6.77 | 11.59 | 1.37 | 4.53 |
| Method | XA($X$)/BJ($Y$) | | | | | | BJ($X$)/XA($Y$) | | | | | |
| | GDP | | Population | | Carbon | | GDP | | Population | | Carbon | |
| | MAE | MAPE | MAE | MAPE | MAE | MAPE | MAE | MAPE | MAE | MAPE | MAE | MAPE |
| Individual | 362.51 | 15.35 | 1102.51 | 6.84 | 374.34 | 8.71 | 500.1 | 18.28 | 1238.58 | 11.92 | 372.03 | 9.23 |
| RP | 230.14 | 7.96 | 699.15 | 3.51 | 215.86 | 3.81 | 224.03 | 7.01 | 795.08 | 7.38 | 213.86 | 6.04 |
| RA | 212.87 | 7.41 | 666.45 | 3.47 | 175.31 | 3.28 | 175.04 | 3.87 | 674.65 | 4.83 | 182.05 | 4.79 |
| HSP | 229.63 | 7.96 | 744.88 | 3.81 | 224.14 | 4.06 | 208.79 | 6.97 | 1042.84 | 10.06 | 306.32 | 8.25 |
| HSA | 167.06 | 5.88 | 630.88 | 3.21 | 186.08 | 3.62 | 188.86 | 5.33 | 735.32 | 6.57 | 227.79 | 6.03 |
| HBP | 264.14 | 8.29 | 905.45 | 4.81 | 261.15 | 5.81 | 207.27 | 6.85 | 963.11 | 9.81 | 291.51 | 7.93 |
| HBA | 170.67 | 6.35 | 659.64 | 3.43 | 191.89 | 3.65 | 173.02 | 3.91 | 682.59 | 4.89 | 181.05 | 4.83 |
| MMD | 242.42 | 8.76 | 856.39 | 4.37 | 236.43 | 4.16 | 198.91 | 6.42 | 788.16 | 6.89 | 231.81 | 6.45 |
| Adv | 235.38 | 8.54 | 886.34 | 5.89 | 260.98 | 5.75 | 241.76 | 8.38 | 972.23 | 9.91 | 257.08 | 6.57 |
| CrossTReS | 187.07 | 7.27 | 674.28 | 3.53 | 196.13 | 3.71 | 188.99 | 4.99 | 740.94 | 6.83 | 217.4 | 6.12 |
| CARPG | 177.08 | 6.71 | 746.81 | 3.84 | 246.33 | 4.41 | 228.84 | 6.98 | 910.22 | 8.67 | 267.8 | 7.32 |
| CoRE | **154.85** | **5.45** | **608.66** | **2.94** | **173.82** | **3.12** | **164.96** | **3.65** | **623.92** | **4.66** | **176.25** | **4.38** |
| Improvement (%) | 7.31 | 7.31 | 3.52 | 8.41 | 0.85 | 4.88 | 4.66 | 5.68 | 7.52 | 3.52 | 2.65 | 8.56 |

- **CARPG** (Yang et al., 2023): Facilitates cross-city knowledge transfer by constructing an inter-city region graph, where regions from the source and target cities are connected based on feature similarity. A reconstruction loss is incorporated to enable joint learning of region representations across both cities during the training stage.

Finally, we introduce a lower-bound baseline, **Individual**, which generates city-specific region representations without any cross-city alignment.

**A.3. Performance Comparison with Baselines**

Table 2 presents the cross-city socioeconomic indicator prediction performance of various methods across XA, BJ, and CD datasets, with $X$ as the source city and $Y$ as the target city. For the four city pairs (BJ/CD, CD/BJ, BJ/XA, and XA/BJ), our CoRE method also demonstrates the best performance.

## A.4. Visualization of Region Representations

We visualize the region representations of two cities by projecting them onto 2D planes using t-SNE (Maaten & Hinton, 2008). The results from the XA and BJ datasets, including CoRE, CoRE w/o LIA (which represents the complete CoRE model excluding the cross-city latent individual alignment component), and the baseline method (HBA), are shown in Figure 8. In these plots, each point represents a region, with green squares for BJ regions and yellow triangles for XA regions. Figure 8 (a) indicates that the region representations from HBA are only partially aligned. Notably, as shown in Figure 8 (b), CoRE w/o LIA exhibits a discernible trend that the overall distributions of region representations from the two cities are similar, suggesting that our latent manifold alignment method enhances the similarity between the data manifolds of the two latent spaces, while still falling short in individual-level alignment. Figure 8 (c) shows that CoRE effectively aligns individual region representations across cities, demonstrating fine-grained alignment of latent spaces from global and local perspectives.

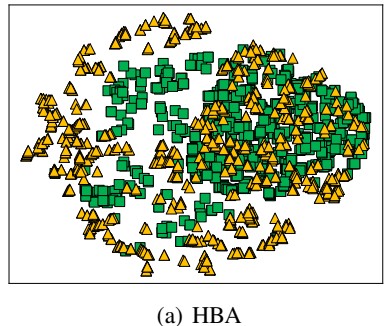 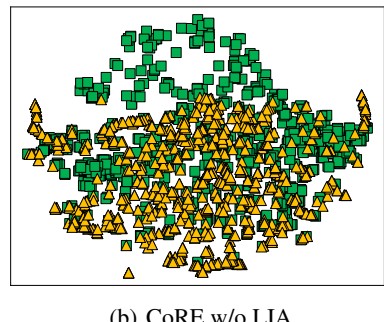 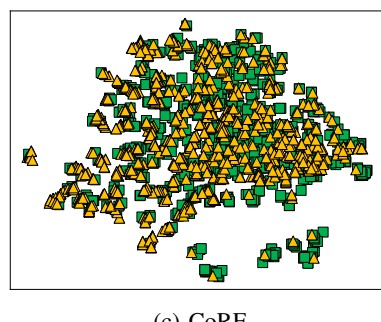

(a) HBA                         (b) CoRE w/o LIA                         (c) CoRE

*Figure 8.* Exhibition of region representations of two cities in 2D planes, with green squares denoting region representations from the BJ data and yellow triangles denoting region representations from the XA data.

## A.5. Within-City Performance

To investigate the potential trade-off between cross-city and within-city predictive performance, we conducted experiments evaluating CoRE on within-city prediction tasks. We compared CoRE against representative mobility-based region embedding methods: HDGE (Wang & Li, 2017) constructs human mobility flow graphs and applies graph embedding to learn region representations; ZE-Mob (Yao et al., 2018) models co-occurrence patterns between regions for representation learning. We also included a variant of our model, CoRE w/o Alignment, which does not perform cross-city alignment and thus serves as a city-specific embedding baseline. We evaluated within-city performance for three cities (XA, CD, BJ) on three downstream tasks: GDP estimation, population prediction, and carbon emission prediction. Results in terms of MAE and MAPE are shown in Table 3.

*Table 3.* Within-city predictive performance on three tasks. "X/Y" denotes training and testing on the same city. The best results per column are in **bold**.

| Method | XA($X$)/XA($Y$) | | | | | | CD($X$)/CD($Y$) | | | | | | BJ($X$)/BJ($Y$) | | | | | |
|---|---|---|---|---|---|---|---|---|---|---|---|---|---|---|---|---|---|---|
| | GDP | | Population | | Carbon | | GDP | | Population | | Carbon | | GDP | | Population | | Carbon | |
| | MAE | MAPE | MAE | MAPE | MAE | MAPE | MAE | MAPE | MAE | MAPE | MAE | MAPE | MAE | MAPE | MAE | MAPE | MAE | MAPE |
| HDGE | 207.36 | 12.58 | 616.37 | 5.45 | 175.24 | 5.95 | 174.39 | 18.98 | 701.82 | 4.97 | 73.84 | 6.08 | 124.55 | 5.57 | 708.58 | 5.57 | 202.60 | 5.43 |
| ZE-Mob | 199.09 | 11.82 | 597.54 | 5.35 | 169.06 | 5.78 | 173.49 | 20.72 | 687.44 | 4.90 | 72.56 | 6.07 | 122.46 | 5.46 | 695.64 | 5.45 | 198.39 | 5.31 |
| CoRE w/o Alignment | **165.36** | **7.24** | **498.38** | **2.96** | **140.81** | **3.31** | **146.35** | **9.05** | **611.24** | **3.01** | **64.34** | **3.41** | **107.13** | **3.24** | **595.11** | **3.08** | **174.82** | **3.19** |
| CoRE | 172.91 | 10.11 | 508.78 | 3.51 | 143.53 | 3.86 | 148.31 | 10.09 | 621.58 | 3.65 | 65.88 | 4.35 | 107.82 | 3.44 | 608.95 | 3.38 | 176.29 | 3.36 |

As shown in Table 3, CoRE achieves strong within-city performance, consistently outperforming mobility-based baselines. CoRE exhibits a marginal compromise compared to its variant without cross-city alignment, which is an expected trade-off for its cross-city alignment capabilities.

## A.6. Ablation Study

The results of CoRE and its variants in terms of MAE and MAPE for the four city pairs (CD/BJ, BJ/CD, BJ/XA, XA/BJ) are shown in Figure 9. We observe that CoRE w/o LMA, CoRE w/o LIA and CoRE w/o One-Stage underperform compared to

the full CoRE model, which further validates the effectiveness of the newly designed cross-city alignment components and one-stage learning paradigm.

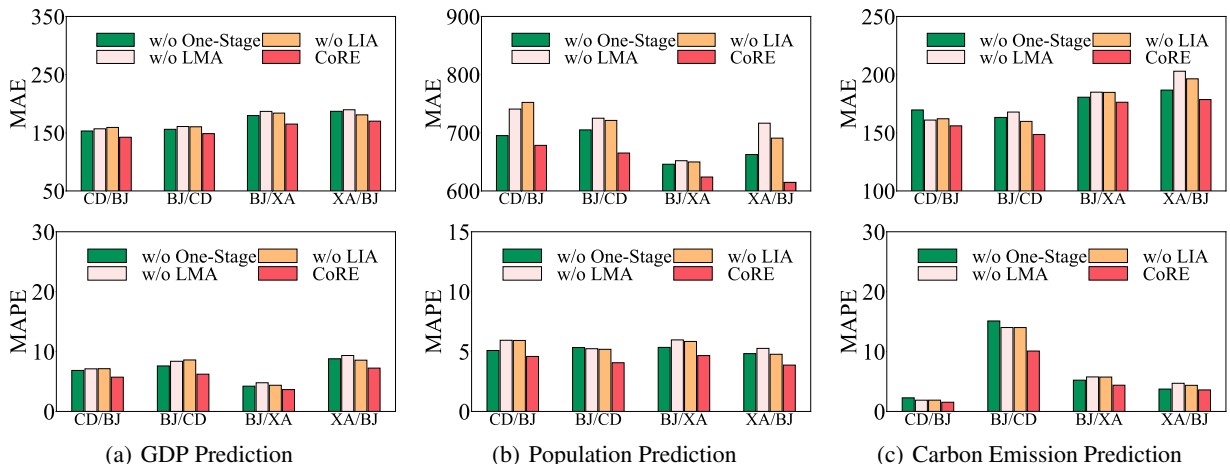

*Figure 9.* Performance comparison of different variants.

