# OpenReview forum: "Cross-City Latent Space Alignment for Consistency Region Embedding"
_ICML.cc/2025/Conference — ICML 2025 poster_

### Official Review · Reviewer_WP7Z · 2025-03-06

**Overall Recommendation:** 4

**Summary:**

This paper deals with a critical issue in the popular trend in urban computing, namely the cross-city latent space alignment problem in region representation learning, which  extracts useful features from different types of urban data for many urban prediction tasks. The issue is that although the pre-training process is free from labels of tasks, doing the actual predictions still need a substantial amount of labels. Oftentimes the labels are not really accessible, so this erodes the principle motivation of region representation.
The paper developed a method that can work without labels, by migrating patterns accumulated from another city to a city with no labels. A fundamental idea here is that the latent embedding spaces of the two cities for pre-trained can be aligned in an unsupervised way without ground truth and rules. The work achieved this by looking at both the overall distribution alignment and pairwise relationship preservation. This is overall an interesting and novel idea to address the critical issue in urban computing.

**Claims And Evidence:**

The overarching claim is the proposed one-stage method for both region representation learning and cross-city alignment can transfer the knowledge learned from a data-rich city to another one where labels are unavailable. The evidence is ample for this claim, including experimental results in multiple tasks, cities as well as baselines. The ablation such as “without one-stage” is convincing.
The other claim is that only looking at embedding distributions rather than using human developed rules are sufficient. This claim is well backed by the experiments too, and it is encouraging to see the advantages of this method with little intervention.
The figure8 for visually presenting the aligned and unaligned latent spaces is useful.

**Essential References Not Discussed:**

I do not find such related works.

**Experimental Designs Or Analyses:**

The experiments are sound with valid tasks, datasets and evaluation metrics. The ablation is well designed with key components replaced or abandoned. For example, abandoning one-stage strategy yields less performance. Parameter sensitivity test is generally well done and makes sense.

**Methods And Evaluation Criteria:**

The method is novel and can be one of the first unsupervised methods on this topic. The method simultaneously learns city-specific region embeddings, cross-city distribution matching and individual level matching. The idea is intriguing and solid.
The evaluation criteria uses public datasets and well established metrics for this problem at hand.

**Other Comments Or Suggestions:**

There are some typos in the paper, and it needs carefully proofreading.

**Other Strengths And Weaknesses:**

1) A promising and sound method is proposed for overcoming a critical challenge in urban computing, which has not been fully explored.
2) The unsupervised method is novel without the need for data pairs in different cities and manual rules.
3) Solid experiments across cities and tasks.
A point that is missing in the paper is what forms a good pair of cities for alignment. If the two cities are extremely different, will this method still work? This is worth further discussing.

**Questions For Authors:**

Refer to weakness.

**Relation To Broader Scientific Literature:**

In an even broader sense, there are increasing works in self-supervised pre-training in urban contexts. Some of them claim to be foundation models. Nonetheless, it is obvious that many of them only focus on specific cities - pre-training in multiple cities is scarce. This submission then ushers an interesting path: overpasses can be established for sharing the knowledge accumulated in different cities. This is interesting in a broader  foundation model sense.

**Theoretical Claims:**

The theoretical foundation of the paper primarily lies in the one-stage method where three components are simultaneously learnt – city-specific region embeddings, cross-city distribution matching and individual level matching. The theories underlying are solid. I have checked mathematical expressions of the model and training objective and found they are error free.

---

> ### Author Rebuttal · Authors · 2025-03-31
>
> **Response to W1:**
> We appreciate the reviewer’s insightful question regarding the suitability of city pairs for knowledge transfer. At this stage of research in unsupervised cross-city alignment, we acknowledge that quantitatively defining the ideal criteria for city pair selection remains an open challenge. Testing numerous city pairs to identify patterns is non-trivial, and quantitatively characterizing a city’s traits—given the multidimensional nature of urban data—adds further complexity. While such analysis is beyond the scope of this paper, we agree it is a valuable direction for future work.
>
> To demonstrate the robustness of our method even when transferring knowledge between highly dissimilar cities, we conducted additional experiments using three cities of China and New York City (NYC). Due to data availability, we evaluated carbon emission prediction (the only task with ground truth in both cities), with results of CoRE and the two best baselines  (HBA and HSA) summarized below:
>
>
> ||XA / NYC||NYC / XA||
> |:-:|:-:|:-:|:-:|:-:|
> |Method|MAE|MAPE|MAE|MAPE|
> |HBA|358.82|4.01|1274.9|59.47|
> |HSA|441.67|4.7|1646.09|66.75|
> |**CoRE**|**279.69**|**2.43**|**514.49**|**16.53**|
>
> ||BJ / NYC||NYC / BJ||
> |:-:|:-:|:-:|:-:|:-:|
> |Method|MAE|MAPE|MAE|MAPE|
> |HBA|325.18|3.64|531.04|19.52|
> |HSA|390.21|4.93|852.23|29.82|
> |**CoRE**|**220.83**|**1.44**|**356.71**|**7.65**|
>
> ||CD / NYC||NYC / CD||
> |:-:|:-:|:-:|:-:|:-:|
> |Method|MAE|MAPE|MAE|MAPE|
> |HBA|239.23|1.39|398.92|38.19|
> |HSA|280.35|2.15|601.38|49.17|
> |**CoRE**|**211.42**|**0.99**|**269.82**|**22.17**|
>
> The results show that CoRE maintains strong performance despite significant differences between cities of China and NYC, reinforcing its adaptability across diverse urban contexts. We acknowledge that further validation across more international city pairs would strengthen these conclusions, and we identify this as an important direction for future research.
>
> **Response to W2:** We sincerely appreciate the reviewer’s careful reading and valuable feedback. We will perform a round of grammar and consistency checks to further improve the clarity and readability of the paper in the revision.

---

> > ### Comment · Reviewer_WP7Z · 2025-04-04
> >
> > I have carefully read the rebuttal of the authors. Some of my concerns are addressed and I will keep my score as accept.

---

> > > ### Author Response · Authors · 2025-04-04
> > >
> > > Thank you very much for taking the time to review our rebuttal. We sincerely appreciate your decision to accept the manuscript and will carefully incorporate your valuable feedback to further improve it.

---

### Official Review · Reviewer_bm5L · 2025-03-06

**Overall Recommendation:** 4

**Summary:**

This paper addresses a critical question in urban computing: Can we align the latent spaces of different cities to leverage knowledge from one city for analyzing others? To tackle this issue, the authors introduce a one-stage Consistency Region Embedding method (CoRE), which combines region embedding learning with cross-city latent space alignment to generate compatible and comparable region representations. Additionally, they achieve cross-city alignment via two interdependent pathways: latent manifold alignment and latent individual alignment, which address both global and individual perspectives. Finally, the authors present comprehensive experiments across three downstream tasks using datasets from three cities.

**Claims And Evidence:**

They claim that the proposed CoRE method ensures compatibility and comparability of region representations from different cities within aligned latent spaces. The experimental results provide evidence to support their claims.

**Essential References Not Discussed:**

Related works are enough.

**Experimental Designs Or Analyses:**

This paper evaluates the proposed CoRE model across three cross-city urban prediction tasks using real-world datasets and compares its performance to various baselines. Table 1 and Figure 5 present the performance comparison results for the CA and CD datasets, along with findings from the ablation study and parameter analysis. The results are comprehensive and effectively support the authors' claims.

**Methods And Evaluation Criteria:**

They achieve cross-city alignment of latent spaces through two interdependent pathways: latent manifold alignment and latent individual alignment. In the latent manifold alignment component, they create virtual parallel common anchors in both spaces, serving as bridges to align the data manifolds based on relative representations. Meanwhile, in the latent individual alignment component, they implement a cross-city attention pipeline that transfers pairwise region correlations between the two spaces, ensuring embedding consistency at the individual region level. The proposed method is not only innovative but also effectively addresses the challenges of cross-city alignment, demonstrating its value and potential in urban computing.

**Other Comments Or Suggestions:**

See strengths and weaknesses.

**Other Strengths And Weaknesses:**

Strengths:
1)	The paper introduces CoRE, a novel framework that integrates region embedding learning with cross-city latent space alignment. This approach addresses the limitations of existing methods by enabling knowledge transfer across cities without relying on hand-crafted correspondence rules, which is a significant advancement in urban computing.
2)	The authors conduct extensive experiments across three downstream tasks (GDP, population, and carbon emissions prediction) using real-world datasets from three cities. The results demonstrate that CoRE outperforms state-of-the-art baselines, showcasing its effectiveness in cross-city prediction tasks.
3)	The paper not only provides a theoretical foundation for cross-city latent space alignment but also offers practical insights by eliminating the need for manual inter-city correspondence rules. This makes the method broadly applicable to unsupervised latent space alignment tasks in urban analytics.
Weaknesses:
1) The paper focuses on socioeconomic predictions using three metrics (GDP, population, and carbon emissions). Expanding the evaluation to include more diverse benchmarks, such as traffic prediction or land use analysis, across different urban science settings would further demonstrate the robustness of the proposed method.

**Questions For Authors:**

Could you extend the evaluation of CoRE to other urban science tasks, such as traffic prediction, beyond the current focus on socioeconomic indicators (GDP, population, and carbon emissions)? If not, explain the reason.

**Relation To Broader Scientific Literature:**

This paper addresses the challenge of aligning latent spaces across cities. On one hand, they align data manifolds of two spaces using the idea of relative representations; on the other hand,  they ensure precise alignment of individual regions across cities through a cross-attention mechanism. The proposed approach eliminates the need for manual inter-city correspondence rules, enhancing its applicability to unsupervised latent space alignment tasks.

**Theoretical Claims:**

I have reviewed them and found no obvious errors.

---

> ### Author Rebuttal · Authors · 2025-03-31
>
> **Response to Q1:** We acknowledge that evaluating the proposed method across additional tasks would further strengthen its empirical validation. Following your suggestion, we conducted supplementary experiments on traffic flow prediction, using datasets from Xi'an (XA) and Chengdu (CD). Specifically, we trained the predictor using labeled data (i.e., region visit counts derived from taxi trajectories) and pre-trained region representations from one city (XA or CD). The trained predictor was then directly applied to predict traffic flow in the other city (CD or XA). The results are summarized below:
>
> ||XA / CD||CD / XA||
> |:-:|:-:|:-:|:-:|:-:|
> |Method|MAE|MAPE|MAE|MAPE|
> |HBA|148.62|6.42|155.53|11.47|
> |HSA|150.01|5.96|149.18|10.18|
> |**CoRE**|**143.79**|**5.12**|**127.59**|**8.46**|
>
> As shown, CoRE consistently outperforms the baselines, reinforcing its effectiveness. While we recognize that no single study can comprehensively demonstrate superiority across all urban computing scenarios, we believe the three primary tasks in our paper, along with this additional cross-city traffic prediction experiment, provide substantial evidence of our method's robustness.

---

> > ### Comment · Reviewer_bm5L · 2025-04-03
> >
> > The author's responses have addressed my concerns. I will keep my rating.

---

> > > ### Author Response · Authors · 2025-04-04
> > >
> > > Thank you very much for taking the time to review our rebuttal. We truly appreciate your positive feedback on our work. We will improve the manuscript based on your valuable feedback.

---

### Official Review · Reviewer_kYEv · 2025-03-07

**Overall Recommendation:** 3

**Summary:**

**Problem:**
- This paper studies the region representation learning problem, with a special focus on cross-city learning
- The cross-city representation learning problem is important because some cities have abundant labelled data, while many others do
- Existing methods rely on heuristic translation functions to align disjoint embedding spaces
    - These typically require hand-crafted rules and require the definition of similar pairs in the two cities, X and Y
    - This is inherently a two-stage approach where translation is disconnected from learning

**Solution:**
- The authors propose CORE, which is a region embedding method that both learns and aligns semantic embedding space across cities
CORE has three key modules:
    - A self-supervised module to learn the primitive embedding spaces of city X and city Y.
    - A latent manifold alignment module that aligns the entire representation space across X and Y using anchor points and reconstruction loss
    - A latent individual alignment module that tries to align the represents of semantically similar regions across X and Y. They do this through cross attention (X→Y and Y→ X) and reconstruction of the empirical covariance of the region embeddings: $Z^X (Z^X)^T$.

**Claims And Evidence:**

The authors primarily make claims around improved cross-city alignment. They construct experiments to test this hypothesis. Additionally, the authors perform an ablation to study to verify the marginal contribution of each of the components they introduce.

**Essential References Not Discussed:**

In addition to the missing baselines above, the authors could consider discussing other early works on region embeddings:

- Jenkins, Porter, et al. "Unsupervised representation learning of spatial data via multimodal embedding." Proceedings of the 28th ACM international conference on information and knowledge management. 2019.

**Experimental Designs Or Analyses:**

Yes, discussed above. Missing baselines

**Methods And Evaluation Criteria:**

As mentioned above, the authors perform experiments to test the cross-city alignment hypothesis. They compare primarily to different variants of Yabe et al. 2020, which constitutes the majority of the baselines. Additionally, they compare to a few domain adaptation methods.

I believe that other baselines should probably be included in their evaluation. Here are some recent examples:

- Yang, Guang, et al. "CARPG: Cross-City Knowledge Transfer for Traffic Accident Prediction via Attentive Region-Level Parameter Generation." Proceedings of the 32nd ACM International Conference on Information and Knowledge Management. 2023.
- Jin, Yilun, Kai Chen, and Qiang Yang. "Selective cross-city transfer learning for traffic prediction via source city region re-weighting." Proceedings of the 28th ACM SIGKDD Conference on Knowledge Discovery and Data Mining. 2022.


Other cross-city learning methods that do require representation learning could also be considered:
- Yao, Huaxiu, et al. "Learning from multiple cities: A meta-learning approach for spatial-temporal prediction." The world wide web conference. 2019.

**Other Comments Or Suggestions:**

See thoughts about figure 7. I'm happy to see hyperparameter study but this figure is not useful.

**Other Strengths And Weaknesses:**

**Strengths:**
- The alignment modules are well explained and are well motivated
- The ablation study is very useful for evaluating marginal impact
- The improvement over the baselines studied indicates efficacy

**Weaknesses:**
- The authors test whether or not CORE can effectively perform cross-city prediction. However, I do think an obvious question is if this comes at the expense of within-city predictive performance. See below for additional thoughts/questions
- Missing baselines
- Figure 7 is not useful. The 3D barchart is very difficult to read or detect differences between hyperparameter settings. I would suggest using heatmaps instead

**Questions For Authors:**

The authors test whether or not CORE can effectively perform cross-city prediction. However, I do think an obvious question is if this comes at the expense of within-city predictive performance. For example, in Table 1, the authors test the cross-city prediction setting of XA(X) → CD(Y) and CD(X) → XA(Y). The performance on this task is quite good (although there are likely additional baselines to try). My biggest concern is what happens to the individual city representations when predicting on data within that same city. For example, after using CORE, what is the predictive performance of GDP_X using Z_X and GDP_Y using Z_Y. Do these representations degrade? How do they compare to the larger body of region representation research?  This is an important question and would warrant further study.

Figure 2: Do you really mean GDP or do you mean income levels? Usually GDP is computed at state or national levels and is not tracked at low levels of spatial resolution.

Table 1: The authors perform five trials and compute average MAE and MAPE over those trials. Why do they not present estimates of the standard deviations of these same statistics? I see they do some type of t-test; it would be helpful to get more details on how that t-test is constructed.

**Relation To Broader Scientific Literature:**

The spatial data sparsity issue likely has many applications to areas of social science, civil engineering or environmental science

**Theoretical Claims:**

No theoretical claims

---

> ### Author Rebuttal · Authors · 2025-03-31
>
> **Response to Q1\&W1:**  We appreciate the reviewer raising this question about the trade-off between cross-city and within-city performance. To thoroughly investigate this aspect, we conducted new experiments comparing CoRE's within-city performance against established region embedding methods using human mobility data: HDGE [1] constructs flow graphs and applies graph embedding to learn region representations; ZE-Mob [2] models co-occurrence patterns between regions to learn representations. While we acknowledge that some recent methods incorporate multi-source data (e.g., POIs, imagery), we focused our comparison on mobility-based approaches due to data availability constraints in our experimental setup. We tested within-city performance (XA→XA, CD→CD, BJ→BJ) with three down-tasks. We also included our proposed preliminary region embedding model (named CoRE w/o Alignment, Section 3.1). The results in terms of MAE are presented below:
>
> |||XA/XA|||CD/CD|||BJ/BJ||
> |:-:|:-:|:-:|:-:|:-:|:-:|:-:|:-:|:-:|:-:|
> |Method|GDP|Population|Carbon|GDP|Population|Carbon|GDP|Population|Carbon|
> |HDGE|207.36|616.37|175.24|174.39|701.82|73.84|124.55|708.58|202.60|
> |ZE-Mob|199.09|597.54|169.06|173.49|687.44|72.56|122.46|695.64|198.39|
> |CoRE w/o Alignment|165.36|498.38|140.81|146.35|611.24|64.34|107.13|595.11|174.82|
> |CoRE|172.91|508.78|143.53|148.31|621.58|65.88|107.82|608.95|176.29|
>
> The experimental results demonstrate that CoRE maintains strong within-city performance, with only marginal performance degradation compared to city-specific embeddings (CoRE w/o Alignment). Additionally, CoRE outperforms mobility-based baselines. More importantly, although it is true that adding the alignment module comes with a marginal sacrifice of within-city performance, the gain of our CoRE alignment method is something that cannot be overstated. It tackles a fundamental challenge in region representation that the embeddings are of little use without task-specific ground truth data. CoRE tackles this problem by aligning the embeddings across multiple cities, and thus enabling predictor reuse.
>
> [1] Region representation learning via mobility flow, CIKM.
>
> [2] Representing urban functions through zone embedding with human mobility patterns, IJCAI.
>
> **Response to Q2:** In Figure 2, we refer to regional GDP levels, not income. These GDP estimates are derived from a high-resolution dataset [3] that provides GDP values at a 1km × 1km grid scale. To adapt this data to our study: we mapped the gridded GDP values to our regions, aggregated the values by proportionally summing GDP within each region, and classified regions into three economic tiers based on these aggregated values. This approach allows us to analyze GDP variations at a finer spatial resolution than traditional state- or national-level metrics.
>
> [3] Forecasting China’s GDP at the pixel level using nighttime lights time series and population images. GIScience \& Remote Sensing.
>
> **Response to Q3:** We appreciate the reviewer's question regarding our statistical analysis. We did indeed calculate standard deviations for all metrics. For instance, the MAE standard deviations of our CoRE for the CD/XA pair were 6.03, 19.72, and 3.34 across the three tasks. Due to space constraints in the table layout, we chose not to include these values in the original submission.
>
> The paired t-tests were conducted as follows: 1) Computing performance differences between our method and each baseline for each of the 5 trials, 2) Performing two-tailed significance tests on these differences, 3) Testing the hypothesis that the mean difference equals zero [4]. The consistently significant results (p < 0.01) across all comparisons provide strong evidence that CoRE's improvements are statistically meaningful, not due to random variation. We will add these details to the revised manuscript.
>
> [4] Hull, David. Using statistical testing in the evaluation of retrieval experiments. SIGIR.
>
> **Response to W2:** We additionally conducted comparisons with CrossTReS and CARPG. While we considered the approach by Yao et al., we found it unsuitable for our unsupervised setting as it requires labeled data in both source and target cities. The results of CrossTReS, CARPG, and CoRE in terms of MAE are presented below:
>
> |||XA / CD|||CD / XA||
> |:-:|:-:|:-:|:-:|:-:|:-:|:-:|
> |Method|GDP|Population|Carbon|GDP|Population|Carbon|
> |CrossTReS|205.48|767.26|153.38|208.17|776.82|148.05|
> |CARPG|190.41|653.27|136.02|186.83|641.94|143.88|
> |**CoRE**|**178.93**|**620.15**|**127.45**|**170.25**|**580.98**|**136.56**|
>
> The experimental results demonstrate that CoRE outperforms these baselines. Similar results are observed in the other four city pairs. These extensive comparisons further validate CoRE's effectiveness in cross-city knowledge transfer. We will incorporate these results into the revised manuscript.
>
> **Response to W3:** We will replace 3D barcharts with heatmaps in the revision.

---

> > ### Comment · Reviewer_kYEv · 2025-04-02
> >
> > I appreciate the additional insights provided in the rebuttal. In particular, the within-city result sheds further light into the behavior of CoRE. I would encourage the authors to include this discussion in subsequent manuscripts. Additionally, the comparison against recent methods (CARPG and CrossSTreS) further validates the claims in the paper.
> >
> > After reading the authors' response I raise my score a 3.

---

> > > ### Author Response · Authors · 2025-04-03
> > >
> > > Thank you very much for taking the time to review our rebuttal. We sincerely appreciate your decision to raise the score. We will incorporate the within-city results and new baselines into the manuscript based on your valuable feedback.

---

### Official Review · Reviewer_Hufa · 2025-03-13

**Overall Recommendation:** 4

**Summary:**

This paper tackles the challenge of transferring urban region embeddings across cities. Instead of the commonly used two-stage approach—first learning city-specific embeddings and then mapping among them—the authors propose a unified one-stage framework called CoRE. The idea is to learn region embeddings from human mobility data while aligning the latent spaces of different cities in one joint process.

In this paper, two key mechanisms are designed: one to embed regions from different cities into distinct latent spaces, and another to align their latent space manifolds while ensuring fine grained compatibility. Experiments conducted on datasets from three cities show that CoRE consistently outperforms several strong baselines on different downstream tasks.

**Claims And Evidence:**

This paper has three major claims: the first is that a unified framework works better than previous two stages works, which ensures compatibility and comparability of region representations from different cities within aligned latent spaces; the second is that cross-city alignment of latent spaces can account for the complex interplay between cities without relying on explicit anchoring pairs of region representations across cities based on human-crafted rules; the third is that by properly aligning the latent spaces, task predictors trained in a data-rich city can be directly applied to another city with minimal degradation in performance.

This paper’s claims have been supported by very well designed and deployed experiments in my opinion. The experimental results clearly indicate that CoRE outperforms various baselines as shown in section 4.2.2, including methods based on hand-crafted anchoring pairs, unsupervised translation techniques, and standard domain adaptation approaches. The improvements in MAE and MAPE are not only consistent across city pairs but are also statistically significant (section 4.3). The ablation study proves that either the manifold alignment or the individual alignment module components are crucial for the overall performance (section 4.4). Visual analysis using t-SNE plots demonstrate that region embeddings from different cities become more tightly clustered and semantically consistent after alignment, which visually supports the authors’ claims (appendix A.4). The downstream tasks are not only on one issue, but on three very distinguished tasks, such as predicting GDP, population, and carbon emissions, showing the very strong capability of the paper applying on different domains for cross city knowledge transfer (Table 1 and Table 2).

Overall, I am very positive with the claims that the authors claimed in the paper, and they have been shown very good evidence to support them. I am very satisfied with both the claims and evidence part.

**Essential References Not Discussed:**

I think that the authors have provided enough references.

**Experimental Designs Or Analyses:**

I have explained my thoughts on experimental designs and analysis in both Claims and Evidence and Methods and Evaluation Criteria. Overall I believe the experiment section is good enough to support the claims of the paper. I am satisfied with this part.

**Methods And Evaluation Criteria:**

The authors start by constructing a mobility graph using human mobility data and then apply a graph attention network (GAT) to learn the initial embeddings. This choice is well justified as GATs are effective at capturing complex relationships in spatial data (section 3.1.2). To bridge the gap between the latent spaces of different cities, the authors introduce a set of virtual parallel common anchors. By computing relative representations of each city’s region embeddings with respect to these anchors, they align the global structure of the latent spaces using an L2 loss (section 3.2). Beyond aligning the overall structure, the method also leverages a cross-attention mechanism to ensure that pairwise correlations between regions are maintained across cities (section 3.3). This cycle-consistency approach forces the fine-grained structure of the embeddings to be preserved.

The proposed CoRE is evaluated on predicting socioeconomic indicators (GDP, population, carbon emissions), which are realistic and meaningful tasks in urban analytics (Table 1 and Table 2).  The use of MAE and MAPE is standard and appropriate for regression problems like these. The authors compare CoRE against a variety of strong baselines, including rank-based methods, hierarchical approaches, and domain adaptation techniques (Section 4.2.2 and section 4.2.3). This comprehensive set of comparisons adds credibility to the results. The paper also includes experiments that explore the impact of the number of virtual anchors and the weights for the alignment losses (section 4.4.1 and section 4.4.2). This analysis helps demonstrate the robustness of the approach.

Overall, I believe that authors have done well with the methods and the evaluation section. The evaluation sections support the claims from the methods in different ways and I think that it is good enough to support all the method claims.

**Other Comments Or Suggestions:**

See Other Strengths and Weaknesses

**Other Strengths And Weaknesses:**

Overall, I think that this is a strong paper. The paper is well written, the algorithms are very well and thoroughly designed, and this paper is on a very interesting topic. I have been enjoying reading the paper. I have explained my thoughts in detail in my previous comments.

Some further suggestions:

The paper used the data all from the cities in China. I am very curious to see how the algorithm would perform under cross-country cities (e.g., NYC to Beijing). Some experiments on this part would be very interesting.

Figure 3 information is a bit too much, took me a lot time to understand this part.

Why in Figure 6, population results are different from GDP and carbon results? Some explanation is needed for this part.

**Questions For Authors:**

None

**Relation To Broader Scientific Literature:**

I can see the potential of the paper’s impact not only limited in the ML application domain, but in a broader machine learning domain especially on the questions related to Latent Space Alignment. Latent Space Alignment is a common problem in many fields, for example, in e-commerce recommendation of different categories. The paper’s proposed methods could be deployed in other ML application fields with similar backgrounds with ease.

**Theoretical Claims:**

Overall, I am satisfied with the theoretical part. Some minor questions:

1. In equation 1, how the graphs of Gx and Gy are constructed in city x and y, this part is not clearly explained in the paper. One or two sentence explanation would be helpful for readers to understand this part.

2. In section 3.3 the destination region rj given ri, what are the meanings of rj and ri here?

3. How will equation 7 be calculated? Will the transpose be calculated first or later?

4. Equation 11 is a commonly used attention mechanism, the authors do not need to spend time explaining this part

---

> ### Author Rebuttal · Authors · 2025-03-31
>
> **Response to W1:**
> We appreciate the reviewer's valuable suggestion regarding cross-country evaluation. To demonstrate the generalizability of our method, we conducted additional experiments using data from three cities of China and New York City (NYC) - cities with significantly different urban characteristics. Due to data availability constraints, we evaluated performance on carbon emission prediction (the only task with available ground truth in both cities). The results of CoRE and the two best baselines (HBA and HSA), measured by MAE and MAPE, are presented below:
> ||XA / NYC||NYC / XA||
> |:-:|:-:|:-:|:-:|:-:|
> |Method|MAE|MAPE|MAE|MAPE|
> |HBA|358.82|4.01|1274.9|59.47|
> |HSA|441.67|4.7|1646.09|66.75|
> |**CoRE**|**279.69**|**2.43**|**514.49**|**16.53**|
>
> ||BJ / NYC||NYC / BJ||
> |:-:|:-:|:-:|:-:|:-:|
> |Method|MAE|MAPE|MAE|MAPE|
> |HBA|325.18|3.64|531.04|19.52|
> |HSA|390.21|4.93|852.23|29.82|
> |**CoRE**|**220.83**|**1.44**|**356.71**|**7.65**|
>
> ||CD / NYC||NYC / CD||
> |:-:|:-:|:-:|:-:|:-:|
> |Method|MAE|MAPE|MAE|MAPE|
> |HBA|239.23|1.39|398.92|38.19|
> |HSA|280.35|2.15|601.38|49.17|
> |**CoRE**|**211.42**|**0.99**|**269.82**|**22.17**|
>
> Our findings show that CoRE maintains strong performance even when transferring knowledge between these geographically and culturally distinct cities. This suggests our method's robustness in handling cross-country urban computing tasks. We acknowledge that further validation across more international city pairs would strengthen these conclusions, and we identify this as an important direction for future research.
>
> **Response to W2:** We thank the reviewer for the comment. Figure 3 shows how we align the hidden data spaces of two cities (like Chengdu and Xi’an) by using a few common anchor points as reference. Instead of comparing raw values directly, we compare how each data point relates to the anchors — like checking how far each region is from landmarks. First, we calculate these relative similarities in each city, and then use those to build a new space where everything is described based on those anchor relationships. We do this again to create a second, more refined version of that space. Finally, we align the two cities by minimizing the difference between their refined spaces, helping the model learn structural similarities even when the cities have different data distributions. I hope this comment can help you clarify the question.
>
> **Response to W3:** We thank the reviewer for the insightful comment. The reason the population results look different from GDP and carbon in Figure 6 is mostly because population patterns are naturally more uneven and complex across urban areas. People tend to live in places based on various personal and social factors—like job locations, housing affordability—which creates irregular and less predictable spatial distributions. GDP and carbon emissions usually reflect clearer, more structured patterns tied to specific economic or industrial areas. Because of this complexity, population prediction has relatively more fluctuation and benefits greatly at first from adding more anchors. In contrast, GDP and carbon emissions data don't need as many anchors to achieve stable predictions, which explains their relatively smoother performance trends when increasing N_a.
>
> **Response to minor questions:**
> We thank the reviewer for pointing out areas that could benefit from further clarification. First, regarding how the graphs are constructed for each city: the mobility graphs are created using human mobility data, where each node represents a region within the city. Edges between nodes represent how frequently people travel from one region to another. These frequencies are normalized based on the total number of trips originating from each region. Second, in Section 3.1.3, the terms r_i and r_j refer to urban regions. Specifically, r_i is the origin region and r_j is the destination region in a recorded trip. The model is trained to predict how likely people are to travel from one region to another based on these patterns. Third, for the computation described in Equation 7, the transpose of the region data is calculated first. After that, the model performs a matrix multiplication with a set of common anchor points. This operation generates a table that shows how similar each anchor is to every region in the city, helping align data across different cities. We will revise the manuscript to clearly explain these parts, so future readers can understand them more easily.

---

### Decision · Program_Chairs · 2025-05-01

**Decision:**

Accept (poster)

**Comment:**

The paper proposes a method to align latent representations between different cities in an unsupervised manner within a unified framework, enabling cross-city knowledge transfer. The method learns representations for each city and then aligns their latent spaces to ensure both global and local consistency. This alignment allows representations to be reused across cities for downstream tasks, which is validated through experiments showing that the proposed method outperforms related approaches.

The reviewers agree that the paper makes a significant contribution to urban modeling, noting that: 1) the core idea is clever and solid, addressing limitations of previous work; 2) the experiments are well-designed, sound, and convincingly support the claims, and 3) the paper is well-written, clearly explaining all aspects. A few concerns were addressed in the rebuttal. I therefore recommend acceptance and encourage the authors to consider the feedback and update the final version of the paper accordingly.